# Clinical Manifestations of Body Memories: The Impact of Past Bodily Experiences on Mental Health

**DOI:** 10.3390/brainsci12050594

**Published:** 2022-05-03

**Authors:** Antje Gentsch, Esther Kuehn

**Affiliations:** 1Department of Psychology, General and Experimental Psychology, LMU Munich, 80802 Munich, Germany; a.gentsch@lmu.de; 2Institute for Psychoanalysis, Psychotherapy and Psychosomatics (IPB), 10557 Berlin, Germany; 3Institute for Cognitive Neurology and Dementia Research (IKND), Otto-von-Guericke University Magdeburg, 39120 Magdeburg, Germany; 4German Center for Neurodegenerative Diseases (DZNE), 39120 Magdeburg, Germany; 5Center for Behavioral Brain Sciences (CBBS), 39106 Magdeburg, Germany; 6Hertie Institute for Clinical Brain Research (HIH), 72076 Tübingen, Germany

**Keywords:** somatic symptoms, psychosomatic, trauma, emotion, hippocampus, insula

## Abstract

Bodily experiences such as the feeling of touch, pain or inner signals of the body are deeply emotional and activate brain networks that mediate their perception and higher-order processing. While the ad hoc perception of bodily signals and their influence on behavior is empirically well studied, there is a knowledge gap on how we store and retrieve bodily experiences that we perceived in the past, and how this influences our everyday life. Here, we explore the hypothesis that negative body memories, that is, negative bodily experiences of the past that are stored in memory and influence behavior, contribute to the development of somatic manifestations of mental health problems including somatic symptoms, traumatic re-experiences or dissociative symptoms. By combining knowledge from the areas of cognitive neuroscience and clinical neuroscience with insights from psychotherapy, we identify Clinical Body Memory (CBM) mechanisms that specify how mental health problems could be driven by corporeal experiences stored in memory. The major argument is that the investigation of the neuronal mechanisms that underlie the storage and retrieval of body memories provides us with empirical access to reduce the negative impact of body memories on mental health.

## 1. General Introduction

We all enjoy remembering moments of bodily comfort, such as a recent massage or a hug from a friend. However, we also memorize negative bodily experiences such as past violence or moments of anger or fear. Conceptually, body memory is defined as the sum of all past bodily experiences that are stored in memory and influence behavior. This entails tactile, motor, proprioceptive, painful and interoceptive experiences as well as accompanying emotions. Body memories therefore comprise corporeal experiences of the past that can be explicit but also implicit in perceptual and behavioral dispositions. Particularly when body memories are implicit [1], they are not easily accessible to conscious reflection. It may be difficult to verbalize our bodily experiences in day-to-day living or even during important life events; nevertheless, they may impact our bodily states at any given moment in time, and particularly during emotional experiences. To understand the influence of body memories on clinical manifestations of psychosomatic disorders or in somatic traces of traumatic life experiences, we need to develop new ways to conceptualize the relationship between body memories and mental health, and we also need to develop new ways to measure body memories empirically.

We here combine knowledge from the areas of cognitive neuroscience and clinical neuroscience with insights from psychotherapy to develop novel concepts on how specific mental health problems could be driven by corporeal representations stored in memory. We explore the specific role of body memories for the occurrence of mental health problems that show ‘somatic’ manifestations, and discuss involved neuronal networks. We argue that better understanding the cognitive and neuronal mechanisms that underlie these processes helps to get empirical access to these phenomena, which allows their scientific investigation, early detection and controlled modification via intervention. We discuss the consequences of our hypothesis for both cognitive neuroscience research and clinical practice as well as for present conceptualizations of mental health problems and their evidence-based investigation.

Given the breadth of the subject, this article does not aim at reviewing all mental health phenomena that could potentially be related to stored corporeal experiences of the past; rather, we will select specific mental health problems for which theoretical and/or empirical knowledge exists for drawing first conceptual relationships between memory mechanisms of bodily experiences and ‘somatic’ manifestations with associated mental health problems. It is also not our aim to argue that other disease models or conceptualizations that are used to describe and treat mental disorders are invalid, or need to be altered. Our aim is to present here a new conceptualization where future research will help to gain supporting or conflicting evidence in reference to present disease models.

Historically, memories of stressful, painful or traumatic bodily experiences have been associated with various mental health conditions. Already the founder of psychotherapy, Siegmund Freud, assigned an important role to memories and in particular the repression of distressful memories that often present in the form of bodily symptoms and co-occur with mental disorders [2]. In his psychoanalytic theory, however, Freud set his focus on intrapsychic reality (unconscious phantasies and inner conflicts) instead of external (traumatic) reality. The impact and re-enactment of physical traumatic experiences within the body was particularly noted by the neurologist and psychoanalyst Sandor Ferenczi (see [3]), and it was in the mid 20th century that Maurice Merlau-Ponty and other philosophers emphasized the importance of body memories as storage units of traumatic life experiences [1,4]. Their view was that, whereas the ‘body at this moment’ can be accessed via ad hoc sensory experiences, the ‘habit body’ can only be accessed via manifested memories of the past [4]. This account offers parallels to the concept of ‘intercorporeal memory’ that specifically investigates the role of bodily experiences encountered during social interactions on our behavior [1]. According to this concept, in particular memories of our earliest bodily interactions with others, for example, the way we are held and comforted by our primary caregivers, shape our interpersonal relations later in life. In a yet broader approach, recent embodied views on human memory emphasize that many of our everyday experiences, such as autobiographical memories, are stored in an embodied format (see [5] for review). In the present article, however, we will not focus on forms of body representation and embodiment per se, but we will discuss how past bodily experiences are encoded, stored and may reappear as somatic manifestations of mental health problems.

From the field of neurology, research on patients with ‘asymbolia for pain’ provides first insights into the neuronal mechanisms that underlie the storage and retrieval of body memories. Patients with asymbolia for pain are sensitive to the perception of pain, but show higher pain tolerance and higher pain endurance compared to control participants. It has been suggested that neuronal disconnections between the somatosensory and limbic system networks give rise to the disorder [6,7]. More specifically, in one study, all patients with asymbolia for pain showed lesions to the insular cortex due to stroke or closed head injury, and one patient even showed an onset of asymbolia for pain after a damage to the posterior insula and adjacent parietal operculum [6]. The insula cortex is regarded as the connection hub between somatosensory experiences and associated memories stored in the medial temporal lobe (MTL) including the hippocampus [8]. In addition, the insula connects painful experiences to feelings of unpleasantness via its projections to the amygdala. Hence, two reasons why patients with asymbolia for pain may show higher pain tolerance and higher pain endurance are that they do not remember past painful experiences associated with certain stimuli due to the disconnection between sensory cortices and the MTL and/or that they show reduced emotional responsivity during painful stimulations. Both could explain why some of them show more risky everyday behavior that may even threaten bodily safety (patient 1 in [6]).

Moreover, research on patients with hippocampal damage provides astonishing evidence for an MTL-mediated ‘forgetting’ of emotional bodily reactions. Patients with hippocampal damage show, for example, no cortisol response during a standardized stress test (public speaking [9]) that usually evokes an increase in cortical levels. Patients with hippocampal damage also spend less time in ‘safe places’ within a virtual environment, and behave less cautiously over time compared to controls, despite explicit knowledge of the threat level [10]. These altered reactions can be observed in spite of preserved amygdala-mediated fear memory (procedural memory, see [11] for review), which indicates that emotional bodily reactions such as stress responses or avoidance behavior are partly based on hippocampal circuits.

As briefly outlined above, historic evidence from the fields of psychotherapy, philosophy and cognitive neuroscience exists for the critical importance of stored bodily experiences for mental health, and clinical examples have provided insights into involved neuronal networks such as sensory cortices including the insula, and limbic systems including the hippocampus. Nevertheless, the detailed cognitive and neuronal mechanisms of how body memories may contribute to ‘somatic’ components of mental disorders are so far largely unknown. Questions such as ‘May somatic symptoms be a manifestation of retrieved past bodily experiences?’, ‘Are negative life events stored in the form of body memories, and how can we alter them?’ and ‘Are bodily dissociations an attempt to ‘forget’ negative bodily experiences of the past?’ open up novel research areas that may help to conceptualize and treat some mental health conditions more effectively, and to recognize them earlier. In particular, this approach allows the transfer of recent insights from the field of episodic memory research, in particular research on limbic system plasticity, to the field of mental health, which may allow the development of novel routes for empirical investigations and intervention.

In this Opinion Article, we will introduce and discuss the hypothesis that the storage and retrieval of bodily experiences offers a critical route to understand and modify mental health problems, in particular bodily trauma, psychosomatic and chronic pain, dissociative symptoms and general somatic symptoms. We will provide empirical evidence to this argument via linking common clinical symptoms to their potential origin in corporeal memories, and will discuss involved neuronal networks and underlying cognitive mechanisms (Section 2.1, Section 2.2, Section 2.3 and Section 2.4). Finally, we will provide an overall summary of how body memory mechanisms may contribute to mental health problems that we introduce as Clinical Body Memory (CBM) mechanisms (Section 3.1, see also Figure 1), and we provide an outlook how these insights can be used to induce plasticity in stored body memories, which may aid therapeutic interventions and prevention in the future (Section 3.2).

## 2. Clinical Body Memory (CBM) Manifestations

### 2.1. Trauma

Trauma memory is perhaps the most obvious example of a clinical and therapeutic concept that is used to describe dysfunctional body memories. Trauma memory is used by clinical psychologists and psychiatrists to describe the encoding of past traumatic life events involving fear of death and other serious bodily threats that can cause the most disturbing and persistent impressions on body memory. Traumatic body memories are particularly observed in posttraumatic stress disorder (PTSD) with intrusively re-experienced traumatic life events that manifest in the form of somatic flashbacks including physical sensations such as smells, tastes, pain, haptic experiences, pressure or sweating. For example, the past experience of pain under torture may reappear in a conflict situation corresponding exactly to the body parts that where affected by the torture [1]. Bauer reports the case of a 37 years old male patient with sudden pain below the armpit extending to his hips, which appeared for the first time many years after the original traumatic event. During this traumatic event, he had experienced electrical torture on both sides of his trunk during political imprisonment [12]. While his physical injuries had long healed and he was otherwise healthy, following a particularly stressful social conflict situation, his body memory seemed to produce pain symptoms from this past traumatic experience. Another example of traumatic body memory is that, after a road traffic accident, people may develop somatic symptoms and panic with particular sensations or sounds, such as ambulance sirens or a weather condition, resembling the physical sensations and circumstances during the actual traumatic event. Similarly, victims of rape during sleep may always awake at the time when the assault took place [1,13].

One clinical concept that has been used to explain trauma memory is re-enactment. It has been proposed that a key characteristic of traumatic memories is that the past is not represented as declarative knowledge (that is, explicitly recollected through images or words) but is re-enacted at the somatic level through immediate bodily experiences and ways of acting [1]. In children, for example, re-enactment of a frightening or painful event has been observed as a frequent consequence of trauma [14]. Very young children in the preverbal period already appear to replicate traumatic bodily experiences in their behavioral actions, in symbolic play and in associated bodily symptoms, even years after the original trauma [15]. Such re-enactment of preverbal trauma has been described, for example, as a form of ‘spontaneously’ triggered internal mental simulation and imagery mediated by mirror neuron networks [14]. Neuroimaging studies have shown that imagining or observing caress-like touch recruits similar neural mechanisms involving the insular cortex as compared to the direct experience of the same kind of affective touch [16,17]. Within a mental simulation framework, trauma memories are therefore assumed to be stored and experienced via the sensory modalities through which they were originally perceived.

According to psychoanalytic theory, it is the ‘unrepresented’ past in particular (‘undigested facts’ from unprocessed somatic experiences [18]) that is replicated in bodily symptoms and re-enacted in interactions with other people. This may occur, for example, in the therapeutic relationship (also called transference relationship) with the psychotherapist, where some of the behavioral patterns are shown towards the therapist during the course of therapy [19]. According to this view, the ‘unrepresented’ past refers to mental states that have not been subjected to internal processes of symbolization or mentalization [18,20] that usually help to attribute meaning to embodied memories. The involuntary mental re-enactment of the trauma may therefore be an attempt to find meaningful linkages between perception, emotions and thoughts in order to prevent re-exposure to the real traumatic event.

The concept of re-enactment is related to the neurocognitive mechanisms of predictive coding, that is, the process of building internal mental models of bodily experiences. According to the free energy principle or the ‘Bayesian Brain’ framework, all adaptive organisms seek to reduce surprise and uncertainty [21]. They do so by inferring the causes of bodily experience using information from previous experiences that are memorized and stored in the form of internal predictive models [22]. Following this line of thought, mental functioning and mental health depends on the ability to continuously update these internal causal models based on the body’s external sensations (exteroceptive, e.g., touch) and the body’s internal sensations (interoceptive, e.g., hunger, pain, cardiac functions [23]). Prediction errors function then as learning signals via revising and refining these models, and via driving actions (active inference [24]) to further minimize errors and improve internal predictive models. According to this view, novel (and yet ‘unexplained’) bodily events either trigger actions to confirm past experiences (i.e., re-enactment) or they are used to test new sensory predictions (i.e., monitoring of bodily signals) in order to improve insufficient causal models and finally achieve homeostatic regulation [25]. In other words, re-enactment may serve the development of accurate internal predictive models of bodily experiences in cases where causal explanations are still insufficient or nonexistent (i.e, ‘unrepresented’ information, see above). Particularly in close emotional relationships, this process is supported by embodied interactions with other people (e.g., mother, therapist, partner) that help to assimilate and give meaning to associated feeling states. If this is successful, it may gradually lead to establishing mental models as well as the capacity for mentalization of bodily experiences. This can reduce uncontrolled bodily re-experiences given they are no longer ‘unrepresented’ [26,27].

Within this framework of predictive coding, the re-experience and expression of body memories in trauma-related disorders may therefore serve the improvement of internal models and the drive to minimize surprise and uncertainty regarding the body in the world. Recall of past experiences is thought to generate prediction errors that train internal models. However, the strong negative and existential impact of psychological trauma may result in overweighting past experiences [28]. As a consequence, trauma-related ‘hypotheses’ (i.e., models) are given a high prior probability, which causes difficulties in performing model updates and incorporating novel bodily experiences associated with specific situations into stored memories. Moreover, triggers that lead to vivid re-experiences of body memories and strong affective states provide further sensory evidence for internal models of the old traumatic event.

With respect to the associated neuronal networks, the hippocampus is thought to facilitate both memory recall and the online processing of predictions via its involvement in MTL-mediated memory mechanisms [29]. In this respect, it is interesting that cognitive neuroimaging studies have revealed a role of the hippocampus in PTSD. Reduced hippocampal volume has been related to trauma, PTSD and major depressive disorder [30,31,32,33], particularly in the subareas CA3 and the dentate gyrus [34,35]. Stevens et al. investigated hippocampal and amygdala function in 54 trauma-exposed women using functional magnetic resonance imaging (fMRI) [36]. Principal component analyses of questionnaire items assessed in a much larger sample were used to define five dimensions of the disorder (negative affect, somatic symptoms, re-experiencing, hyper-arousal and numbing) with associated principal component weights of each dimension within each participant. Within the MR scanner, participants observed neutral, positive or negative scenes that needed to be recalled 30 min later. This allowed the authors to later measure hippocampal and amygdala activity during the encoding of events that were recalled versus non-recalled. The authors found that hippocampal activation during the encoding of recalled versus non-recalled scenes correlated with principal component weights of the re-experiencing dimension of the disorder. No such association was found for the other four dimensions. In addition, there was an interaction between the principal component of re-experiencing and emotion condition, where the correlation was only observed for the encoding of neutral and negative but not positive scenes that were later recalled. In a whole-brain analyses of the recalled versus non-recalled trials, the authors also showed that re-experiencing was positively correlated with encoding-related activation in the bilateral amygdala, left hippocampus, dorsomedial prefrontal cortex, bilateral inferior frontal gyrus, middle frontal gyrus and lateral temporal cortex. For the other four dimensions, there were no significant clusters in which symptoms predicted encoding-related activation. These data indicate a hyperactivation of the limbic system including the hippocampus during the encoding of neutral and negative scenes in those trauma patients who particularly suffer from re-experiencing negative past memories. These findings provide support for the involvement of the hippocampus in the re-enactment of traumatic body memories.

In addition, there is evidence that the amygdala mediates the effects of emotion on involuntary sensory memory. For example, amygdala stimulation has been shown to trigger vivid emotional bodily memories such as a sudden odor of burnt wood [37]. With respect to traumatic events, dual representation accounts suggest that negative emotions on the one hand strengthen sensory images through up-regulating the amygdala, while they also weaken the association between individual items and their context due to hippocampal down-regulation [38,39]. Thereby, negative affect is thought to boost intrusive negative bodily memories in anxiety disorders, such as PTSD, via different effects on distinct memory systems.

Taken together, traumatic intrusive body memories are a clinical condition that may relate to the retrieval of stored bodily experiences of the past. According to clinical concepts, re-experiencing may be related to memories that are ‘unrepresented’ and unmentalized, and that are re-enacted especially in stressful and conflictual situations. Predictive coding theory similarly assumes a re-experience of past events when causal explanations are missing to optimize model predictions, and assumes that higher weight is given to past experiences of high emotional value. Both predictive coding theories and neuroimaging studies support the involvement of the limbic system including the hippocampus and the amygdala in this process where hyperactivation, reduced model update and enhanced (emotional) recall may be three possible underlying mechanisms relevant for CBMs.

### 2.2. Pain

Other cases where body memories may lead to clinical manifestations are psychosomatic and chronic pain. The widely used term ‘pain memory’ already hints towards the assumed influence of past painful bodily experiences on the ad hoc perception of pain. By definition, pain memory effects encompass the higher subjective perception of pain over time even when the physical pain decreases, but also pain responses in the absence of any apparent physiological origin. Suffering from pain without apparent physiological origin can be observed in patients diagnosed with pain disorder, fibromyalgia, somatization disorder and irritable bowel syndrome. Fibromyalgia, which is characterized by chronic widespread musculoskeletal pain, also belongs to the group of functional somatic syndromes, and often co-occurs with somatoform disorders, where the etiology of this condition is not well understood [40]. In addition, there is a high comorbidity and there are shared risk factors between chronic pain, major depressive disorder and anxiety disorders [41]. Estimates in the USA assume that between 30 and 50 million citizens suffer from chronic pain [42], which is a significant burden both for affected individuals and society as a whole.

One clinical condition that has been used to study pain memory is phantom limb pain. Around 80% of all patients with an amputated limb experience pain from the region of the amputated limb [42]. Among different mechanisms that have been proposed to underlie this phenomenon, such as (the lack of) cortical reorganization after amputation, compensation [43,44] or peripheral nerve damage [45], sensory memory traces are also frequently discussed [42,46]. Sensory memory traces are expected to form when intensive states of pain cause an enhanced representation of pain in primary somatosensory cortex (SI) topographic maps and associated brain areas, which increases their responsivity to the processing of pain also after amputation. This is supported by the observation that the subjective perception of pain after amputation is similar to pain experiences before amputation [46]. This would be similar to the case of chronic back pain patients where an enlargement and hyperactivity of the area in SI that represents the back has been reported [47]. It has therefore been suggested that ‘somatosensory pain memories manifest themselves in alterations in the SI cortex and may contribute to hypersensitivity even in the absence of peripheral stimulation’ [42].

In addition, it has been noted that the posterior insula and other brain networks that are involved in pain processing mediate pain relief and activity changes that represent the phantom hand [48]. In a meta-analytic and ‘transdiagnostic’ approach, common structural brain changes were identified that characterize chronic pain, major depressive disorder, as well as anxiety disorders [41]. The authors found that decreased gray matter volume in the bilateral insula, dorsomedial prefrontal cortex, bilateral anterior cingulate, supplementary motor area, superior temporal gyrus and lateral prefrontal cortex characterized all three disorder categories, confirming an important role for modulatory networks but also the insula for the occurrence of mental health problems. However, common functional network changes that characterize all three disorders could not be identified, leaving open the question whether the same or different functional mechanisms are associated with the structural brain changes.

The assumption of altered sensory and insula networks in chronic pain is supported by a study where sensory-evoked potentials in response to painful stimulation were compared between fibromyalgia syndrome patients and controls using electroencephalography (EEG) [49]. Even though the intensity of the painful stimulation was adjusted to the 50% threshold for all participants, fibromyalgia syndrome patients showed higher N80 amplitudes compared to healthy controls. Higher N80 amplitudes likely reflect increased processing of pain in sensory cortices, the insula and the anterior cingulate cortex in these patients. It has been argued that this increased responsiveness towards painful stimulation results in a heightened salience of painful percepts [40], where hyperexcitability may be a central mechanism [50]. This is in line with an fMRI study on patients with fibromyalgia. The authors found that in response to painful stimulation, functional connectivity between the affected SI leg area and the bilateral anterior insula was increased in fibromyalgia patients, but not in healthy controls [51]. The functional connectivity between the SI leg area and the anterior insula additionally correlated with clinical/behavioral pain measures and autonomic responses.

A similar finding was reported by an intervention study using non-invasive neurostimulation. The authors used a double-blind and sham-controlled design, and applied anodal transcranial direct current stimulation (tDCS) to the sensorimotor cortex of amputees suffering from phantom limb pain while they were performing phantom hand movements [52]. The authors show that a single session of non-invasive neurostimulation significantly relieved phantom limb pain in the intervention group, and that these effects lasted for at least one week. Interestingly, fMRI analyses showed that pain relief was associated with reduced activity in the sensorimotor cortex of the missing hand area after stimulation, and that pain relief and reduced sensorimotor activity correlated with preceding activity changes during stimulation in the mid- and posterior insula and the secondary somatosensory cortex. Given the importance of the insula cortex in connecting sensory cortices to the MTL (as outlined above for the case of ‘asymbolia for pain’ patients, see Section 1), these studies indicate an important role of altered sensory-limbic coupling for pain memory and phantom limb pain.

It is worth noting, however, that changes in the SI topographic map architecture in chronic pain patients have not always been reported. Mancini et al. used fMRI to investigate patients with complex regional pain syndrome affecting the hand and control participants while they received tactile stimulation at the fingers [53]. The authors did not find any difference of the affected hand representation in patients with respect to cortical area, location and geometry when compared to control areas and control participants. The authors also did not find a relation between clinical metrics of the disorder and SI topographic map characteristics. However, the connectivity between the affected SI area and the insula was not investigated here, and tactile rather than painful stimulation was provided to the hand. The mechanistic involvement of primary sensory representations in chronic pain may therefore depend on precise disease characteristics, chosen stimuli and the etiology of the disorder.

Another neurocognitive mechanism that has been used to explain chronic pain is predictive coding (see Section 2.1 for an introduction to the concept). Here, the idea is that individuals with chronic pain have heightened predictions of experiencing pain due to past experiences of pain stored in memory, which results in heightened pain perception even during harmless bodily sensations [54]. Following this view, pain perception emerges because pain is inferred as the most likely cause for sensations where no other causes can be found. Reduced model update and enhanced recall of prior experiences of pain are therefore the assumed underlying mechanisms. This explanation is similar as for cases of traumatic body memories discussed above where biases in perception due to overweighting of past experiences are assumed. According to this view, missing information is searched in memory (particularly in the hippocampus) and false percepts are filled in to reduce sensory uncertainty. However, it is worth noting that the causes of phantom limb pain and the importance of central and peripheral factors still remain to be clarified [55].

Taken together, there is initial evidence that psychosomatic and chronic pain may partly be explained by sensory memory traces, for example, in the form of altered topographic maps, sensory cortex hyperexcitability and hyper-/hyposensitivity. In addition, increased sensory-limbic coupling via the insula may be involved in the development of pain sensations without apparent cause, and may be important also for modulating CBM mechanisms. In spite of this initial evidence, more research is needed to investigate the precise and mechanistic relationships between the subjective perception of pain and pain experiences stored in memory.

### 2.3. Dissociation

The clinical concept of somatoform dissociation refers to physical symptoms reflecting a bodily disconnection, that is, a certain detachment and separation from one’s own body. Non-pathological forms of body dissociation include absorption in activities and retreating into fantasy such as daydreaming. More severe pathological forms of disconnection from the body are observed in traumatized individuals who report feeling unreal, numb or robotic, leading in extreme cases to depersonalization disorder with chronic self-fragmentation and disturbance of identity. In patients with dissociative disorder, a phenomenological distinction has been drawn between psychoform dissociation (such as disruptions of memory and identity) and somatoform dissociation (i.e., somatic symptoms affecting sensorimotor functions, with no apparent physical cause [56,57]). Dissociative somatic symptoms suggest a physical defect or dysfunction and may include sensory losses (e.g., deafness, blindness, gustatory and olfactory alterations), analgesia (e.g., hyposensitivity for pain), kinesthetic anesthesia (numbing), perceiving missing body parts even if the body is intact or loss of control over bodily functions (e.g., not being able to swallow, difficulty urinating or paralysis). They may be associated with acute or chronic presentation in psychiatric disorders including PTSD, Borderline Personality Disorder and somatoform disorders. The somatoform dissociation questionnaire (SDQ-20 [58]) measures the severity of such bodily experiences.

It is well-known that dissociation affects and is affected by memory. According to the dissociative encoding hypothesis, peritraumatic dissociation (occurring at the time of trauma) affects the encoding of trauma-related experiences leading to a general increase in physical symptoms and somatization [59,60]. It is assumed that a failure to integrate sensory memory traces into declarative memory may leave (implicit) body memories intact [38] causing, for example, intrusive flashbacks (see Section 2.1 above) or fragmentary recall of memories. Van der Hart et al. investigated self-reported memories in dissociative identity disorder patients and observed abnormalities in basic memory processing also for non-traumatic events [61]. In particular, memory recall in these patients occurred as a somewhat detached somatosensory experience and in the form of sensory fragments (e.g., vivid smells, tastes or somatic sensation), often lacking a clear autobiographical narrative or any related narrative during initial recall. In this view, dissociation appears to affect memory. Conversely, however, memory mechanisms may also lead to dissociative symptoms. Classical concepts of dissociation in hysteria by Pierre Janet and Siegmund Freud view unresolved or repressed (unconscious) traumatic memories as the cause for a tendency to dissociate (see [62]). According to this so-called defense hypothesis, somatic dissociative symptoms that follow the traumatic event serve to avoid the recall of stressful memories by disconnecting and protecting from unpleasant, overwhelming bodily experiences. Consequently, repressed body memories may have a corporeal presence in the form of somatic ‘blind spots’ and may manifest themselves, for example, in specific sensory losses such as if a ‘part of the body is just gone’ (for a detailed clinical case, see [63]).

A case report of a young woman with left-side conversion (i.e., functional neurological) symptoms involving sensory loss illustrates how a cerebral lesion may cause reactivation of implicit sensory memories, which contributes to the formation of dissociative sensory symptoms [64]. Following a traumatic event of rape, the woman developed physical symptoms such as skin swelling and rashes as well as sensory sensations of numbness confined to the exact left side of her body involving face, neck, trunk and limbs. In neurological examinations, a right parietal infarct was identified using MRI; however, no evidence of true sensory deficits was found using somatosensory evoked potentials. Psychotherapy led to a complete remission of sensory symptoms, which confirmed the diagnosis of somatization disorder with dissociative conversion symptoms. It was proposed that decreased corticofugal inhibition (i.e., inhibition via nerve fibers or tracts that stem from the cerebral or cerebellar cortex) due to parietal lesion permitted reactivation and increased intrusion of somatosensory memories represented at the thalamic level leading to symptoms of dissociative sensory loss. Other studies have confirmed the association between the appearance of dissociative symptoms and the recall of emotionally repressed past stressful life events. For example, Kanaan and colleagues report neuroimaging data from a patient with right-sided paralysis of both upper and lower limbs who lost all sensation bisecting the trunk [65]. The recall of an emotionally repressed event that was crucial to the genesis of her symptoms activated a network centered at the amygdala and the right inferior frontal cortex, when compared to the recall of a different but also severe past life event. While the patient denied the emotional significance of the event, the recall was associated with primary motor cortex deactivation in the affected limb. The authors argued that the emotional experience may have been processed at the physiological level causing the paralysis.

Another possible supporting mechanism underlying the expression of body memories as somatoform dissociation may be a failure to perceive or integrate sensory signals from within the body (i.e., hyposensitivity towards visceral or metabolic signals) that appear to play a central role in both dissociative and memory functions. Impaired interoceptive processing has been suggested to underlie dissociative symptoms in patients with functional seizures [66], dissociative disorder [67] and desomatization in depersonalization disorder [68]. According to a recent framework of interoceptive inference [69], top-down predictions of interoceptive signals evoked by visceral responses to sensory signals influence the degree of dissociation and emotional awareness. Bayesian accounts [70] have suggested that dissociative symptoms are caused by abnormally elevated precision of bottom-up sensory (error) signals, that is, by failures to represent uncertainty. Accordingly, somatic dissociative symptoms may require some attention allocation for their maintenance, reflecting an aberrant attentional bias towards (traumatic) prior beliefs involving the affected body part.

Furthermore, perceived agency, that is, perceived control over the body and environment, is assumed to play a critical role in the generation of interoceptive predictions and dissociative symptoms [69]. The interoceptive predictive coding model assumes that self-awareness and feelings of presence (which are absent in depersonalization symptoms) are determined by the ability to predict the internal and external consequences of actions. Body memories of inescapability and helplessness due to an experienced inability to fight or flee during a traumatic event have therefore been connected to symptoms of dissociative freezing and behavioral immobility [63].

Abnormalities in limbic system function including the hippocampus and the amygdala, as well as the insular cortex, have been well established in the neurobiological literature on dissociative symptoms (both psychoform and somatoform). In particular, hippocampal volume has been shown to be smaller in dissociative identity disorder with PTSD compared to patients with PTSD-only or to matched controls, which was found to correlate negatively with severity of dissociative symptoms and childhood trauma ([71,72], for review see [73]). Furthermore, smaller insular cortex and reduced white matter volume has been repeatedly reported in dissociative identity disorder [73,74]. Interoceptive brain circuits including the anterior insular cortex have also been implicated in emotion memory functions such as threat memory, e.g., [75]. In addition, aberrant formation of interoceptive associations and interoceptive recall has been suggested to contribute to somatic symptoms such as appetite loss in depressed individuals (e.g., [76]). Given the above outlined importance of the insula cortex in mediating the connectivity between sensory and memory circuits, decreased sensory-limbic coupling could also explain the contribution of the insula to dissociative symptoms. However, neuroscientific research is incomplete regarding the relation between dissociative loss of body memories and the development of somatic symptoms in dissociative disorders.

Taken together, dissociative sensory symptoms may be an expression of repressed (traumatic) body memories that are either reactivated only in parts, or that are inhibited due to alterations in sensorimotor and limbic system networks. Aberrant predictions due to a bias towards prior beliefs associated with past bodily experiences, potentially also mediated by attention, have been discussed as a potential cognitive mechanism. Moreover, a failure to integrate interoceptive signals may contribute to the dissociative nature of somatic symptoms associated with functional loss. Similar as for traumatic intrusive body memories and chronic pain, also here, reduced model update may be an involved mechanism.

### 2.4. General Somatic Symptoms

Above, we have discussed the potential involvement of clinical body memory (CBM) mechanisms for mental illness such as for patients with trauma-related disorders, chronic pain syndromes or dissociative symptoms. A more common case of somatic manifestations of past bodily experiences may be general somatic symptoms. General somatic symptoms are here defined as bodily experiences of unknown cause that can appear as part of a clinical diagnosis of somatoform disorder or somatic symptom disorder in the ICD-10 and DSM-5 psychiatric classifications [77,78], respectively, but that can also appear without such a diagnosis. Indeed, it is often the lack of a clear diagnosis that causes problems for people suffering from general somatic symptoms to find appropriate treatment. While the concept of ‘medically unexplained symptoms’ includes somatic symptoms with or without organic origin [79], our definition focuses on somatic symptoms that present without an organic origin as the main causal factor. General somatic symptoms can be considered an unexplained mass phenomenon of Western societies where according to estimations, three quarters of all visits of a general practitioner are due to bodily symptoms where no medical cause can be identified [80]. Since the co-morbidity between somatoform disorders and other psychiatric disorders such as depression and anxiety is high [81], general somatic symptoms, particularly when not associated with a clear diagnosis, are an overlooked and largely neglected phenomenon relevant for health care.

Could some general somatic symptoms, similar to the other clinical phenomena discussed above, be a sensory manifestation of stored bodily experiences? Initial evidence on the role of memory mechanisms for the manifestation of general somatic symptoms stems from research on patients with injuries or surgeries at specific body parts. For example, 20–30% of people who experienced a heart attack suffer from heart symptoms and depression afterwards, even if heart function is intact [82]. Similarly, most people who experienced an injury to the musculoskeletal system have sensations at the affected body part afterwards, where 20% of them meet the criteria for a mental disorder, even if the affected body part has been treated successfully [83]. Moreover, oral symptoms after dental treatment, such as chronic pain or occlusal discomfort, for which the cause remains unknown, are frequently reported. This phenomenon has been referred to as medically unexplained oral syndrome (MUOS) where psychological origins are assumed to be causally involved [84]. In spite of such initial evidence, empirical research that precisely investigates body part-specific encoding and retrieval of bodily experiences and their potential manifestation as general somatic symptoms is so far scarce.

One first step is to consider whether those neuronal and cognitive mechanisms that we have identified above, and which are assumed to mediate CBM storage and retrieval, may also apply (even though perhaps in milder forms) to general somatic symptoms. To start with the limbic system, mechanisms discussed above include limbic system hyperactivation, enhanced recall and reduced model update. Indeed, one recent model on the etiology of medically unexplained symptoms assumes that physical symptoms in general (irrespective of whether an organic cause exists or not) are not direct reports of bodily sensations but an inference based on implicit predictions about interoceptive information derived from prior knowledge [79]. One of the central claims of this model is that the brain interprets signals from the body in the light of predictions that are based on past experience. Medically unexplained symptoms are here understood as a ‘distortion in awareness brought about by the over-activation of symptom representation in memory, with various top-down factors serving to maintain this’ [79]. According to this view, past experiences of injury at the arm, for example, may lead to the perception of pain at the arm after recovery because unusual sensations have the best model fit with pain perception even though the match between the prediction (i.e., the pain) and the input (e.g., tingling) may be low. The result would be that the perception is distorted in the direction of the prior, i.e., the experiences stored in memory. Greater activation of affective networks and reduced inhibition are two suggested mechanisms to explain augmented and imprecise prediction errors that can lead to distorted perception, which may finally lead to somatic symptoms.

At the neural level, one fMRI study compared anxiety patients with healthy controls, and found higher amplitudes of low-frequency fluctuations in the left thalamus and left hippocampus in the patient group compared to the control group, where the connectivity difference was positively correlated with the the 15-item somatic symptom severity scale of the Patient Health Questionnaire (PHQ-15) [85]. Another MRI study used voxel-based morphometry to investigate 288 healthy participants, and found a positive correlation between somatic complaints (measured as the summed score for the self-rating anxiety scale scale composed of 15 somatic and five affective symptoms) and the volume of the parahippocampal gyrus adjacent to the entorhinal cortex [86]. Based on these results, the authors emphasized the importance of the parahippocampal gyrus for memory consolidation and emotional learning, and suggested that increased parahippocampal/entorhinal cortex volume may be associated with increased vulnerability to somatic complaints in the general population.

Even though these studies and theories provide initial evidence for a potential involvement of hyperactivation/reduced inhibition of limbic system networks in relation to general somatic symptoms, anxiety-related symptoms could not clearly be dissociated from somatic symptoms in the above cited literature (due to the investigation of anxiety patients and the use of the self-rating anxiety scale to assess somatic symptoms, respectively). In addition, limbic system activation has not been investigated in a memory task but only during rest. In order to study the relationship between limbic system hyperactivation, enhanced recall and general somatic symptoms in healthy people, future studies should investigate limbic system activation both during memory encoding and retrieval in otherwise healthy participants. In addition, somatic symptoms and anxiety-like behavior should be investigated using independent metrics where the latter should dissociate between state-anxiety and trait-anxiety [87].

In addition to the limbic system, above, we also discussed the possible involvement of sensory memory traces and cortical modulation for the development of CBMs. With respect to general somatic symptoms, one resting state fMRI study compared two patient groups with major depressive disorder (with and without somatic symptoms) and found significant reductions in regional homogeneity and the amplitude of low-frequency fluctuations in the bilateral precentral gyrus, bilateral postcentral gyrus (i.e., SI) and left paracentral gyrus in the somatic group as compared to the pure depression group [88]. Interestingly, also in the above cited study on fibromyalgia where painful stimulation increased the functional connectivity between the affected SI leg area and the bilateral anterior insula in patients but not in healthy controls [51], decreased resting state functional connectivity in the sensory system was reported. Another fMRI study compared depressive patients with somatic symptoms to healthy controls, and found that at baseline, depressive patients showed weaker functional connectivity between the ventral anterior insula and the right orbitofrontal cortex (OFC) than controls. In addition, the strength of the correlation between the ventral anterior insula and the right OFC was negatively correlated with the 15-item somatic symptom severity scale of the PHQ-15 and both depressive and somatic symptoms as assessed with HDRS scores. Interestingly, electroconvulsive therapy reduced depressive and somatic symptoms, and increased functional connectivity between the ventral anterior insula and the right OFC [89]. A drawback of both studies is, however, that depressive symptoms could not be dissociated from somatic symptoms due to the investigation of depressed patients in both studies.

With respect to sensory hyperexcitability, one behavioral study reported reduced tactile thresholds in a vibrotactile detection task in participants suffering from somatoform disorders, which corresponds to a higher tactile sensitivity in this group [90]. It has also been shown by the same group that medically unexplained symptoms relate to higher amounts of false alarms in a vibrotactile detection task both in healthy controls and in individuals suffering from somatoform disorders [90,91]. The authors argue that the Somatic Signal Detection Task (SSDT) they used in their studies offers a paradigm that is helpful to identify processes relevant for somatoform disorders and general somatic symptoms. Whether higher amounts of false alarms can be interpreted as higher or lower sensory sensitivity is, however, debated. It has also been shown that participants and patients with medically unexplained symptoms show lower correspondence between actual respiratory changes and self-reported breathlessness in a CO_2_ inhalation task [92,93], hinting towards lower interoceptive sensitivity. In addition, higher sensitivity and attention towards sensory stimuli in patients with somatic symptoms have also been discussed in the context of physiological arousal and anxiety-like behavior (‘amplification model’ [94] from [79]). According to this view, anxiety and associated physiological arousal increases bodily attention and also increases the associated processing of sensory stimuli leading to their enhanced perception. If sensory hypersensitivity in the context of general somatic symptoms exists, and whether or not it is associated with established sensory memory traces, higher arousal or both, remains to be clarified.

Taken together, there is initial evidence towards a differential involvement of the limbic system, cortical control mechanisms and sensory perceptual systems in relation to the occurrence of general somatic symptoms that show some similarity to CBM mechanisms in clinical conditions outlined above. Nevertheless, basic research studies that dissociate somatic from associated depressive or anxiety symptoms, that investigate symptom development over time and that study the reactivity of cortical networks in conditions of simulation or memory consolidation are so far scarce. Given the high comorbidity between somatic symptoms and mental disorders, and the often untreated cases of mild or preclinical somatic symptoms in the general population, further research is needed to close these and related knowledge gaps.

## 3. Clinical Body Memory (CBM) Mechanisms

### 3.1. Summary and Outlook

Above, we have presented condensed and accumulated evidence from the fields of cognitive neuroscience, clinical neuroscience and psychotherapy that supports the view that the storage and retrieval of past bodily experiences are one causal factor for the development of distressing and persistent bodily symptoms (Clinical Body Memory (CBM) Mechanisms, see Figure 1 for a summary). Bodily experiences include the feeling of touch, pain or inner signals of the body (i.e., interoceptive signals) often in combination with emotional experiences such as stress, discomfort or fear. We provide evidence for the hypothesis that particularly negative body memories, that is, negative bodily experiences of the past that are stored in memory and influence behavior, contribute to the development of somatic manifestations of mental health problems including traumatic re-experiences, chronic pain, dissociative symptoms or general somatic symptoms when their retrieval, updating, cortical modulation and/or sensory manifestations are altered compared to healthy individuals. Mediating neuronal mechanisms were identified in the limbic system (including hyperactivation, enhanced recall and reduced model update), in sensory memory traces (including altered topographic maps, hyperexcitability and hyper-/hyposensitivity), in the insula (including altered sensory-limbic coupling) and via cortical modulatory factors (such as de-/increased inhibition, and attentional biases). We argue that the investigation of these and other CBM mechanisms that underlie the storage, retrieval and modification of body memories will offer novel routes to reduce the negative impact of body memories on mental health (see Figure 1).

Before we discuss the implications of our hypothesis for clinical intervention in more detail, it should be noted that the CBM mechanisms introduced here do not reflect a deterministic view, but underline the importance of multifactorial causation for the etiology of somatic and somatoform symptoms. In accordance with current views on mental health, the psychosomatic complaints described above are the result of a complex interplay of a variety of biological, psychological and sociocultural factors, where body memories may act as one, but not the only, causal factor or trigger. Accordingly, the mechanisms of memory storage and retrieval may offer one potential route towards the development of clinical interventions, but should always be accompanied by other interventions to appropriately treat mental disorders as a multifactorial and multicausal phenomenon. In addition, it is worth noting that the way this Opinion Article is structured reflects a categorization of mental disorders into trauma, chronic pain, dissociative symptoms and general somatic symptoms. While other categorizations would have been possible, ‘transdiagnostic’ views can additionally reveal unbiased insights by detecting common brain changes across distinct disorder categories that may share common mechanisms (see, for example, [41]). Meta-analytic approaches and those that investigate large cohorts without a priori categorization may be a fruitful way to identify common mechanisms of dysfunctional body memories across present disorder categories in the future.

In this article, we have concerned with the relationship between somatic manifestations in mental disorders (and in healthy people) and memory mechanisms by considering the possibility that some bodily symptoms are due to retrieved or inhibited experiences of the past and/or due to changes in sensory processing due to a history of emotional bodily experiences. While the importance of memory mechanisms has also been stressed in prior research and in clinical models on mental health, in particular with respect to trauma patients, the novelty of the CBM concept is to stress the importance of studying the neuronal mechanisms that underlie the storage and retrieval of past bodily experiences to understand their contribution to mental health. The reason why this is critical is that research on episodic memory has in the past mostly focused on visual perception [95,96,97], in particular object perception [98,99,100], autobiographical memories [5,101,102] including emotional memories [103,104,105], as well as everyday experiences such as spatial navigation [106,107], but has so far not focused on the mechanisms of how everyday bodily experiences, for example, driven by touch, pain or interoception, are stored and retrieved by memory networks. The CBM mechanisms summarize evidence to support the view that the investigation of how bodily experiences are stored and represented in memory is crucial in order to develop effective treatments for mental disorders where those mechanisms are causally involved.

In our view, it would therefore be critical to study the role of sensory memory traces and their modulation in more detail to understand to which extent also general somatic symptoms can be explained by retrieved body memories that represent past bodily experiences. Sensory memory traces have been discussed in the context of chronic pain; however, also for general somatic symptoms, altered functional connectivity within somatosensory networks, between somatosensory networks and the insula and altered top-down influences may be influential. Which exact neuronal mechanisms mediate hyper-/and hyperexcitability of the somatosensory systems in the context of long-term memory retrieval is, however, unclear; also the specificity of altered sensory representations (with respect to the affected body part and the stimulus provided) remains to be clarified. It further remains to be tested whether the generalizability of sensory memory traces relates to symptom severity, and whether those mechanisms can be altered by intervention.

With respect to cognitive CBM mechanisms, reduced model update has been implicated in distinct symptomatic expressions of body memories and may therefore be an important transdiagnostic CBM mechanism. Bayesian approaches to mental dysfunction (see, e.g., [108]) emphasize how past experiences may lead to dysfunctional probabilistic representations of bodily sensations that are resistant to change. Negative memories, in particular early adversity, are thought to act as strong prior expectations in internal models of the body operating at all levels of the cortical hierarchy, from sensory observations to abstract, complex beliefs about the causes of illness. As reviewed above, these strong prior expectations are thought to bias attention and interpretation of afferent input in a way that is consistent with past aversive or traumatic experiences, leading to the manifestation of somatic symptoms such as pain. Furthermore, action and, in particular, interaction with others may play an important role in CBM mechanisms. For example, the clinical concept of re-enactment of negative body memories and the notion of active inference both point to the role of action in coping with uncertainty resulting from ‘unrepresented’, unmentalized past experiences or from incomplete, deficient internal causal models. Furthermore, defensive dissociation and memory fragmentation may act as cognitive CBM mechanisms underlying, for example, symptoms of sensory loss. However, the integration of clinical therapeutic concepts and existing basic neurocognitive models warrants more conceptual work, which promises to be a fruitful endeavor for future empirical research on CBM.

We further expect that illuminating the role of the ‘visceral brain’ will provide new insights into the ways body memory affects mental health. For example, advances have been made in studying cortical memory traces of past immune responses, that is, how the brain stores and remembers information about inflammation and specific immune challenges in the body. Koren et al. report a study in mice where a past inflammation in the colon re-emerged after reactivation of insular neurons that were active during the initial inflammation [109]. Their findings indicate that memory alone can activate the immune system in the absence of an outside trigger. In other words, the brain remembered an old infection and generated the disease (i.e., peripheral inflammation) on its own by reactivating a specific memory trace of the past bodily immune response. This and similar recent findings demonstrate how psychosomatic symptoms can be understood and investigated as reactivated body memories of past immune responses. Attenuating memory traces in the insula may be a potential avenue for treating psychosomatic diseases; however, testing these hypotheses warrants further investigation in humans. Furthermore, this study highlights the crucial role of the brain’s insular cortex in body memory. As outlined above, converging evidence from animal and human studies points to the insula as a core hub region for integrating bodily information with memory (especially in the MTL) and emotional content (especially in the amygdala), and for mediating defensive and regulatory behavior. However, despite increasing research on the role of insula for mental health, e.g., [109,110], it still remains a largely understudied brain region in the context of memory research and body memory in particular [111].

In addition, more research is needed to understand the role of emotional modulation and emotional memory for the development of CBMs. There is, for example, evidence that the amygdala differently encodes recognition memory of objects that were encoded under social stress versus more neutral social interactions [112]. In one study, two groups of healthy participants were exposed to either a social stress condition (mock job interview in front of an evaluation committee) or to a more neutral social interaction condition (free talk about career aspirations in a neutral manner), and encountered different objects during this time (e.g., a tea cup). Half of these objects were used by one of the committee members, making them ‘central’ to the episode, the other half of the objects was not used by the committee members (i.e., peripheral objects). Later, both groups were asked to recognize both the central and peripheral objects among distractor objects behaviorally and in an MRI session. The authors found that the group that encountered the social stress condition was superior in memory recall compared to the group that encountered the normal social condition. Using representational similarity analyses, it was also shown that the representations in the left amygdala of central objects that were encountered in a stressful condition were more similar to each other than they were to distractor objects. In addition, the representations of the central objects became more similar to the face of the stressor, which predicted later memory. No such effects were found in the right amygdala, or in the hippocampus. Given this effect was only shown for central but not peripheral objects, these results indicate an important role of the left amygdala for the generalization and increased salience of stressful memories as well as for relating stressful experiences to the social cue inducing it. Alternatively, these results could also reflect the higher similarity of the emotional responses triggered by single items encountered during a stressful episode.

Finally, an aspect that remains particularly critical to solve is the involvement of spatial memory networks in the development of mental disorders. While recent research has highlighted the importance of the MTL not only for encoding distances in spatial navigation, but also for encoding relational distances in other domains, such as object perception [113,114,115] or social relationships [116,117], the role of the hippocampus and the entorhinal cortex in the storage of somatosensory signals remains rather vague. Early theories assume an important role of the ventral pathway that connects the MTL to the secondary somatosensory cortex for establishing somatosensory memories [118], but this and also other pathways remain to be studied in more detail. In this view, CBMs highlight the importance for future research in this area to foster an integrated view of human memory function and its impact on behavior.

### 3.2. Possible Interventions

In this last section, we will draw first conclusions on how interventions could be used to induce plasticity in dysfunctional body memory networks. We here summarize existing evidence on the plasticity of body memories in order to drive forward new ways of developing experimental investigations to reduce some of the symptoms listed above based on the theoretical framework presented in this article.

Training interventions that increase bodily and somatosensory awareness as well as interoceptive skills are one obvious way to change body memories. Training provides novel and repeated experiences to the brain that can potentially be used to alter network functions over time. The importance of sensory training has, for example, been emphasized in relation to the potential role of predictions for the emergence of somatic symptoms [79]. If prior predictions can influence bodily awareness in a way to interpret normal sensations as potentially harmful, then one reason why this occurs is low precision and/or low weighting of actual bodily perceptions in this process (i.e., reduced model update, hyposensitivity, see Figure 1). Training the accurate perception of bodily signals is therefore one way to change existing models towards more realistic ones. In this respect, interoceptive exposure therapy and training people to become more sensitive to different interoceptive sensations are two suggested approaches [79]. For the case of phantom limb pain, extensive training with a prosthesis, intense sensory input to the missing limb area and sensory discrimination training were proposed to be effective in reducing phantom limb pain [42]. In one study, for example, electrical stimulation was used to excite the nerves that formerly supplied the amputated arm. Within a 2-week training session, patients learned to discriminate between the frequency and the location of the stimulation, which induced cortical reorganization and reduced phantom limb pain [119].

Another example that shows the potential effectiveness of training on mental health in the context of body memory is major depressive disorder. Particularly overgeneralization of autobiographical memories is considered to be a critical mechanism underlying the pathogenesis of major depressive disorder (for review see [120]), which has been associated with impairments in social problem solving, executive control, and future thinking (for review see [121]). Memory training is therefore increasingly used to reduce depressive symptoms by changing the underlying stored experiences, and offers one way to transfer insights from cognitive neuroscience to therapy. Because patients with depressive symptoms also sometimes show sensory memory impairments, such as reduced pattern separation abilities [122], one may expect that sensory memory training, where participants learn to more effectively use bodily experience for decision making and bodily control, may also be an effective way to target dysfunctional body memories in some cases.

Psychotherapy, which focuses on narrative processes, emotion awareness and emotion expression related to physical symptoms, and which also targets interpersonal difficulties, has been shown to be effective in severe somatoform disorder [123]. According to the ‘borrowed brain’ model, it is the interpersonal dimension of therapy in particular that can create changes in memory traces by influencing predictive coding processes (see [27]). The psychotherapist helps to generate top down predictions by resonating at the non-verbal level (i.e., facial expression, tone of voice and other embodied gestures and countertransference phenomena, see [19]) and by training the ability of the patient to mentalize (i.e., to understand one’s own and others’ mental states) via reflective discourse. This process equals successful parental mentalizing, which helps the child to integrate ‘unnamed’ feelings into predictive models of bodily states such as hunger and satiation, cold and warmth, pain and relief. This can increase the predictability of the infant’s environment including bodily, emotional and social consequences. The recent concept of ‘embodied mentalization’ [26] provides a mechanistic account of how subjective bodily experiences are formed in development, by being progressively mentalized and understood in interaction with others. Similar to the patient-therapeutic case, this occurs primarily through embodied interactions between parent and child (i.e., so-called shared ‘we-experiences’ [26]). Brief psychodynamic interventions based on the concept of mentalization have already been successfully developed for individuals with functional somatic disorder [124] and general somatic symptoms [125].

Yet another approach is to target the cortical modulating factors to influence stored memories and/or their influence on behavior. Several theoretical concepts exist that assume a powerful influence of mental techniques to control bodily perceptions, which can be performed also without outer assistance or social interaction. For example, autosuggestion and autogenic training can be used to modulate tactile and painful bodily experiences (for review see [126]), and could therefore potentially be helpful in changing ad hoc bodily experiences, such as somatic symptoms, without external help. Autosuggestion is characterized by volitional, active control over one’s own physiological states via the reinstantiation and reiteration of an idea (for example, ‘my arm feels warm’) that counteracts an actual physiological state (for example, ‘my arm hurts’ [126]). This could be used, for example, to overwrite misperceptions where tingling at the arm is falsely interpreted as pain (see Section 2.2 above). However, empirical evidence on whether or not autosuggestion can be used to reduce somatic symptoms or overwrite existing memories is, to the best of our knowledge, so far lacking.

Another potential way to alter CBMs and to study the neural changes induced by therapy is to reactivate a stored memory trace under controlled experimental conditions, and to alter it by incorporating novel experiences. Recent research on human memory reconsolidation suggests that reactivation can induce a labile, unstable state that needs restabilization during which memory can be modified [127]. This could also be a promising avenue for interventions targeting body memory traces following trauma [128]. For example, invasive neuroscientific methods such as transcranial magnetic stimulation (TMS) could potentially help to disrupt and erase trauma episodes from memory [129], whereas bodily cues such as sound or smell could be used during deep sleep stages [130] to stabilize memories in an updated form. One study on women with endometriosis (a chronic and painful disorder of the uterus) applied an intervention that was based on the reactivation of a memory trace. The therapeutic intervention motivated patients to remember negative memories of the past and to report associated bodily symptoms [131]. The therapist then combined the retrieved memories with positive bodily feelings of the affected body part, such as acupuncture or touching the body with warm objects. This approach was reported to be successful in reducing symptoms of endometriosis.

In this respect, it is also relevant that pharmacological intervention after memory reactivation may aid the modification of memories. The beta-adrenergic blocker propranolol, for example, reduces mental imagery of an emotional event when given within six hours of the actual event [132]. One study has tested whether propranolol intake not after the actual event but after memory retrieval of a traumatic event also influences mental imagery of the event thereafter [133]. Participants received either propranolol (*n* = 9) or a placebo (*n* = 10) after scripting parts of their traumatic event. One week later, participants listened to the playing of their own script and imagined the scene for 30 s while physiological responses were measured. Heart rate and skin conductance responses in the intervention group were reduced compared to the placebo group indicating that propranolol intake reduces the vividness of a stored memory after reconsolidation. However, this study did not investigate the effect of propranolol without memory reconsolidation, and the sample size was relatively small, which warrants further clarification on the underlying neuronal mechanisms. In addition, it is worth noting that some studies also report no effect of propranolol intake within 12 h after the actual event on later physiological responses during imagery [134].

Moreover, as already mentioned in Section 2.2, non-invasive neurostimulation techniques may be a promising way to alter affected neuronal networks. In the above-cited study, neurostimulation of the sensorimotor system in amputees while they moved their phantom hand reduced phantom limb pain, and altered sensory-insula networks [52] hinting towards a possible modulation of altered pain memory networks. Moreover, a meta-analysis concludes that evidence for the therapeutic efficacy of repetitive transcranial magnetic stimulation (rTMS) exists, based on the differences in therapeutic efficacy of real versus sham rTMS protocols, replicated in a sufficient number of independent studies [135]. The authors conclude that, among others, level A evidence (definite efficacy) is reached for high-frequency rTMS of the primary motor cortex contralateral to the painful side for neuropathic pain, and for high-frequency TMS of the left dorsolateral prefrontal cortex for depression, whereas Level B evidence (probable efficacy) is reached for high-frequency rTMS of the left primary motor cortex or DLPFC for improving quality of life or pain, respectively, in fibromyalgia, high-frequency rTMS of bilateral primary motor cortex or the left DLPFC for improving motor impairment or depression, respectively, in Parkinson’s disease, and high-frequency rTMS of the right DLPFC in PTSD [135]. This initial evidence hints towards possible successful approaches to use neurostimulation to alter CBM mechanisms.

Finally, an interesting future avenue for research would be to extend recent developments in the area of epigenetics to the field of body memory research. Increasing evidence demonstrates how environmental events modify gene expression thereby contributing to the formation of long-lasting memories [136] and also mediating the effects of childhood trauma on psychopathology [137]. Epigenetic mechanisms such as DNA methylation are thought to influence neuroplasticity causing long-lasting changes in the brain in response to stressful experiences (especially during critical periods of development) which can predispose an individual to mental disease such as PTSD [138]. An emerging body of research suggests that epigenetic-based therapies can promote suppression of memories [139], and such epigenetic modifications may also be produced by psychotherapy [140]. Therefore, studying the neuroepigenetic patterns contributing to CBM mechanisms may aid the development of (preventative) psychological or chemical interventions that alleviate the somatic consequences of negative body memories.

Taken together, the development of clinically relevant reconsolidation-based interventions specifically targeting body memories and related somatic symptoms, and potentially also incorporating pharmacological intervention, therefore remains a major challenge for future research. Virtual Reality (VR) may be a particularly promising new technology in research studies to combine with such approaches, because it allows the creation of realistic, precisely controlled and individualized environments in a laboratory setting. The clinical potential of VR as an embodied technology has been repeatedly demonstrated for the treatment of mental disorders [141]. Moreover, it is increasingly used in cognitive neuroscience research, for example, to study spatial navigation [142], or emotional experiences [143], which could help to reactivate and modify stored memory traces. It will remain a thriving endeavor to transfer these new insights from the laboratory and virtual reality technology to real life and clinical contexts for alleviating the somatic consequences of negative body memories.

## Figures and Tables

**Figure 1 brainsci-12-00594-f001:**
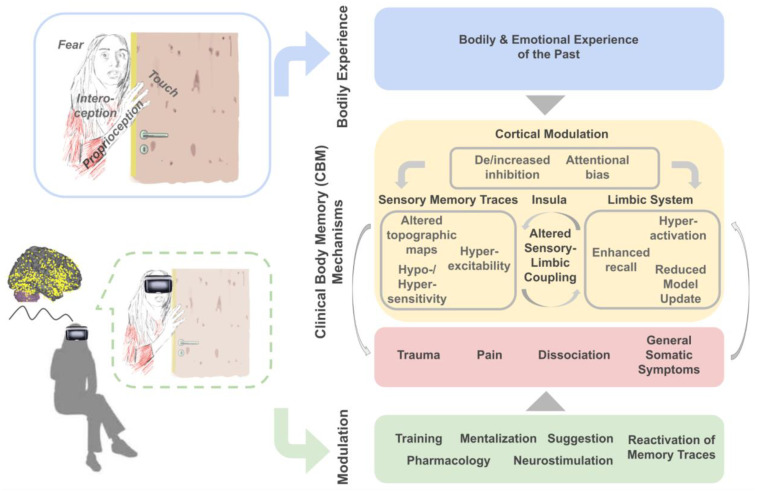
**Clinical Body Memory (CBM) Mechanisms.** The key hypothesis discussed here is that stored bodily experiences of the past and associated emotions (blue boxes) can contribute to the development of Clinical Body Memory (CBM) mechanisms including trauma, pain, dissociation and general somatic symptoms (red box) via neuronal and cognitive mechanisms that mediate their storage and retrieval (yellow box). Experimental investigation may allow empirical access and modulation of CBMs (green box), for example, via using Virtual Reality (VR) paradigms (left bottom).

## Data Availability

Not applicable.

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
