# Peer review of "Clinical Manifestations of Body Memories: The Impact of Past Bodily Experiences on Mental Health"

_brainsci, 2022, doi:10.3390/brainsci12050594_

Round 1
Reviewer 1 Report
This is an exciting and inspiring manuscript. The authors present a good review of cognitive neuroscience, clinical neuroscience, and psychotherapy that supports the hypothesis that the storage and retrieval of past bodily experiences are one antecedent factor for developing of distressing and persistent bodily symptoms. Clinical Body Memories associated would contribute to the development of clinical phenomena like trauma, pain, dissociation, and general somatic symptoms. However, the authors also prevent considering CBM mechanisms in a deterministic view but underline the importance of multifactorial causation for the etiology of somatic and somatoform symptoms. Body memories act as one but not the only causal factor or trigger.
Complementing this review, I celebrate that the authors also evaluate the implications of their hypothesis for clinical intervention in the second part of the manuscript. They discuss the plasticity of body memories to drive forward new ways of developing experimental investigations to reduce some of the symptoms associated with the analyzed alterations. They present some results from the experimental investigation that may allow empirical access and modification of Clinical Body Memories.
They review the approaches like training the accurate perception of bodily signals, which allow changing existing models towards more realistic ones. They also cite other theoretical concepts that attribute a powerful influence of cognitive techniques to control bodily perceptions. One example of the potential effectiveness of training on mental health in the context of body memory is a major depressive disorder. Psychotherapy, which focuses on narrative processes, emotion awareness, and emotion expression related to physical symptoms, has also been effective in severe somatoform disorder. The authors underline Virtual Reality (VR) as a particularly promising new technology to combine with such approaches because it allows the creation of realistic, precisely controlled, and individualized environments in a laboratory setting.
I believe that the manuscript would benefit from including more results from applying a technique such as TMS in depression or techniques such as TDCS in changing some mental associations, inhibiting or exciting some areas.
In the same way, the authors talk about the benefits of reactivating a stored memory trace under controlled experimental conditions and altering it by incorporating novel experiences. It would be interesting at this point to cite also the studies like those using adrenergic inhibitors like propranolol in the consolidation of memories or when reactivating stored traumatic memories (Brunet et al.’s. study: “Effect of post-retrieval propranolol..”; Hoge et al.’s study: Effect of Acute Posttrauma propanolol…). In the same way, I think it would be interesting to mention an up-and-coming venue like mapping the epigenetic changes linked with some mental diseases and how the low-level epigenetic pattern sustaining the problem can be modified through psychological interventions or possible chemical interventions. It would be interesting to extend these new insights to this field to drive possible interventions to alleviate the somatic consequences of negative body memories.
Reviewer 2 Report
The authors proposed the hypothesis that "negative body memories, that is, negative bodily experiences of the past that are stored in memory and influence behavior, contribute to the development of somatic manifestations of mental health problems including somatic symptoms, traumatic re-experiences, or dissociative symptoms". Also, they provided a potential mechanism behind this process, and nice figure for future directions. I really enjoyed reading this opinion article. This article has great implications for future mechanisms and clinical studies.
Author Response
Thank you for your kind review.